# Perceived quality of care and choice of healthcare provider in informal settlements

**Chris Conlan**[1]*, **Teddy Cunningham**[1], **Sam Watson**[2], **Jason Madan**[3], **Alexandros Sfyridis**[1], **Jo Sartori**[2], **Hakan Ferhatosmanoglu**[1], **Richard Lilford**[2]

1 Department of Computer Science, University of Warwick, Coventry, United Kingdom, 2 Institute of Applied Health Research, University of Birmingham, Birmingham, United Kingdom, 3 Warwick Medical School, University of Warwick, Coventry, United Kingdom

* chris.conlan@warwick.ac.uk

## Abstract

When a person chooses a healthcare provider, they are trading off cost, convenience, and a latent third factor: "perceived quality". In urban areas of lower- and middle-income countries (LMICs), including slums, individuals have a wide range of choice in healthcare provider, and we hypothesised that people do not choose the nearest and cheapest provider. This would mean that people are willing to incur additional cost to visit a provider they would perceive to be offering better healthcare. In this article, we aim to develop a method towards quantifying this notion of "perceived quality" by using a generalised access cost calculation to combine monetary and time costs relating to a visit, and then using this calculated access cost to observe facilities that have been bypassed. The data to support this analysis comes from detailed survey data in four slums, where residents were questioned on their interactions with healthcare services, and providers were surveyed by our team. We find that people tend to bypass more informal local services to access more formal providers, especially public hospitals. This implies that public hospitals, which tend to incur higher access costs, have the highest perceived quality (i.e., people are more willing to trade cost and convenience to visit these services). Our findings therefore provide evidence that can support the 'crowding out' hypothesis first suggested in a 2016 Lancet Series on healthcare provision in LMICs.

## Introduction

More than half of all people living in low- and middle-income countries (LMICs) live in cities and, particularly in Africa, these cities are growing rapidly [1]. A large proportion of the urban population live in informal settlements or slums [2]. There is evidence that the quality of primary healthcare available to people in LMICs is of low quality [3,4]. In rural areas, people have little choice of provider due to the distances involved in travel. The situation is different for the urban poor. A recent study [5] shows that people living in informal settlements have access within 40 minutes to a very wide variety of sources, from single-handed practitioners to clinics and hospitals (both publicly and privately funded).

The issue of what types of healthcare people will seek when they need outpatient care is of great importance, as the distribution of different types of healthcare providers (HCPs) is

**Data Availability Statement:** All datasets pertaining to this work are available on request from: https://zenodo.org/record/6783011 (DOI: 10.5281/zenodo.6783011).

**Funding:** This research was funded by the National Institute for Health Research (NIHR) (16/136/87) using UK aid from the UK Government to support global health research. CC and TC are supported in part by the UK Engineering and Physical Sciences Research Council under Grant No. EP/L016400/1. The funders had no role in study design, data collection and analysis, decision to publish, or preparation of the manuscript. RJL is also funded from the NIHR Applied Research Collaboration (ARC) West Midlands. The views expressed in this publication are those of the author(s) are not necessarily those of the NIHR or the UK government.

**Competing interests:** The authors have declared that no competing interests exist.

critical in accommodating healthcare demand–an issue that can be influenced by policy makers [6]. In light of this, a recent Lancet Series was dedicated to the question of identifying variability in the configuration of service provision in terms of the size of the facility and its funding structure [7–10]. The study concluded that "reasonable quality" public services may crowd out informal providers [8] (informal healthcare providers are defined as those that are not state-authorised or registered). However, the evidence for this conclusion was coarse-grained, since it was based on the correlation between aggregate fiscal allocations to public HCPs and aggregate demand. The purpose of this paper is to conduct a finer-grained study to investigate patient choice of facility in urban areas; specifically, whether (and to what extent) people will travel further, incur greater inconvenience and/or pay more to reach preferred healthcare providers. Bypassing is a recognised feature of health-seeking behaviour in LMICs [11–14]. We observe patient bypasses of HCPs in order to determine patients' choices, which can reflect the relative 'perceived quality' of the facilities in the study. The analysis is stratified by the facilities' sizes and funding sources in order to verify the market mechanism behind the 'crowding out' hypothesis.

Leonard et al. investigate bypass behaviour amongst health-seekers in an LMIC setting and define the terms "observable quality" and "unobservable quality" to describe and understand the behaviour. They conclude that "there is strong evidence that patients know about unobservable features of health facilities" and that this contributes towards their decision making when seeking healthcare [14]. A more recent study [5] formalises this notion further, positing three broad factors that influence a person's choice of healthcare provider: (i) monetary cost– such as out-of-pocket expenses to include the costs of travel, the service itself (e.g., consultation, medicines, tests), lost earnings; (ii) inconvenience–including effort and time; and (iii) a latent factor, which is often referred to as 'perceived quality' [12,13,15]. In this study, we observe people's health seeking behaviour with respect to cost and inconvenience in order to provide evidence on perceived quality. McFadden [16] states that the decision (i.e., choice) a person makes can be modelled with the use of choice models that can be populated with hypothetical choices or observed behaviour (i.e., expressed preference). This is thus a study of expressed preferences, rather than hypothetical choices, such as those that might be elicited in a discrete choice experiment [16]. We use this observed behaviour to define a model that considers not just the overt choices (e.g., to visit an HCP), but also the hidden choices (e.g., the HCPs that were bypassed), and aggregate these choices to quantify the notion of perceived quality.

We utilise data covering healthcare providers and households across four slums in two African cities (Nairobi, Kenya and Ibadan, Nigeria). The data was collected as part of a larger study on healthcare provision [17], where surveys of households and visited HCPs were conducted. The surveys collected data on, among other things, transport costs, the time taken to reach the HCP, waiting time, the condition for which care was sought, and healthcare costs. Our data-driven method uses well-known concepts from transportation and geography literature, such as the generalised access cost. We develop an access cost function that incorporates transportation cost, inconvenience, opportunity cost, and the monetary cost of treatment. By applying this function to visits observable in the data, we can determine when a facility is bypassed and the relative magnitude of that bypass with respect to cost and time, and thus calculate a residual term that we call the 'perceived quality index'.

## Methods

### Overview

Using geotagged dwelling and HCP data, and individual-level HCP visit data, we determine the occasions when someone bypasses one HCP to reach another HCP. A bypass is determined

**Fig 1. Summary of research approach.**

using an access cost function, which we refer to as the access cost. As stated above, the access cost describes the cost an individual incurs in order to access healthcare and includes transportation cost, inconvenience, opportunity cost (i.e., loss of earnings), and the monetary cost of treatment. The cost function we use derives from the well-known concept of "generalised access", which combines various costs associated to access into a single measure, and a single unit. Bocarejo et al. [18] review methods for calculating accessibility in an LMIC setting. They note that the generalised access cost has been used in the great majority of relevant studies. A bypass occurs when someone visits an HCP, but an alternative HCP exists with a lower access cost for that individual. As we determine bypassing using the access cost, we are controlling for the monetary and opportunity cost, and inconvenience of a visit, and by aggregating the bypasses across all observed visits we calculate a residual term at the HCP level that we call the perceived quality index (PQI). For example, when one facility is often bypassed in favour of another the visited facility would accrue a high PQI (while the bypassed facility's PQI would lessen) in proportion to the magnitude of difference in frequency of bypass and the two factors that compete with quality: monetary cost and inconvenience. In practice, we observe that people will very often bypass multiple HCPs, and each of these bypasses contributes to the PQI of both the bypassed facility and visited facility.

Our approach is summarised in Fig 1, which shows the input data, processing steps, and the methods we employ towards determining perceived quality. There are assumptions in our approach: the probability that a facility type is bypassed may depend on the symptom constellation and severity; a person may bypass a facility because they do not know it is there; and a visit may not originate at home. We examine these assumptions below and analyse them further in the Discussion.

## Data

For the purposes of our research, we utilise information from two different types of data– survey and geospatial. The data was collected as part of the National Institute for Health Research-funded Global Health Research Unit on Improving Health in Slums, a five-year study to look at healthcare availability, access and use for people living in slums across Africa and Asia [17]. In Table 1, we present a summary of the four selected slums.

**Table 1. Data summary of surveys collected by slum.**

| Slum | Number of Dwellings Surveyed | Number of Inhabitants Surveyed | Number of Facilities Surveyed |
|---|---|---|---|
| Korogocho (Nairobi) | 1508 | 3878 | 66 |
| Viwandani (Nairobi) | 1089 | 2724 | 91 |
| Sasa (Ibadan) | 1304 | 4794 | 100 |
| Idikan (Ibadan) | 849 | 2936 | 79 |

**Geospatial data.** The geospatial data contains information on all structures and the road network within each slum as shown in Fig 2. The spatial dataset (i.e., shapefile) generated for each slum indicates the structures, classified into dwellings where people live and non-dwelling structures, therefore providing locational information on the set of HCPs within the slum boundaries. The underlying road and footpaths network (RFN) for each slum, and the surrounding area, was extracted from OpenStreetMap (OSM) [17,19] under the Open Database Licence.

**Survey data.** Surveys were conducted for each slum in the form of a questionnaire on an individual and HCP level. In the case of individuals, the survey comprised 506 questions and focused on the respondents' socioeconomic circumstances, as well as their relationship to, and interactions with, HCPs–described in detail elsewhere [5,21]. The individual-level survey was conducted on thousands of individuals for each slum, as detailed in Table 1. Each respondent's last visit to an HCP's outpatient department and the details pertaining to this visit (e.g., the reason for the visit, the associated costs, and the time taken to reach the HCP) were recorded. The HCP-level survey contained questions pertaining to the type of facility, the services available, and its administration, such as source of funding.

**Healthcare facilities.** The set of HCPs was compiled in two steps. First, we include all HCPs within the slum boundaries that were identified from the geospatial data collection, and all HCPs included in the HCP-level survey [17]. Second, we include some of the most frequently visited HCPs mentioned in the responses of the individual-level survey not captured earlier; these additional HCPs were mainly located outside of the slum boundary. The additional HCPs were catalogued manually by our in-country partners, who located the sites on Google Maps and derived basic information such as facility type and funding source from secondary sources, such as Google and/or their local expertise.

We conduct our analysis in two dimensions: by funding source (i.e., public vs. private), and by facility size (i.e., clinic vs. hospital). Accordingly, all HCPs are classified into one of four classes: private clinics, public clinics, private hospitals, and public hospitals. Note that the original HCP survey collected data on other HCPs, such as dentists, maternity clinics, and pharmacies. However, we only conduct the analysis for HCPs that we consider to be reasonable substitutes for one another as they provide general primary care ('polyclinic') services. For example, if someone bypasses a dentist to visit a maternity clinic, this choice is driven by clinical need, rather than perceived quality, and we therefore exclude this type of HCP.

We assume that hospitals and clinics are comparable HCPs in terms of presenting symptoms/condition. To verify this assumption, we compare the distribution in patient needs across clinics and hospitals (see Table A in S2 Text), and we find these profiles to be very comparable. We discuss these findings further in S2 Text. It is possible, or likely, that even among a condition type, severity varies by chosen HCP, and we discuss this point more in the Discussion.

**Ethics statement.** All participants provided informed consent to participate before taking part in the study. The NIHR Global Health Research Unit on Improving Health in Slums was

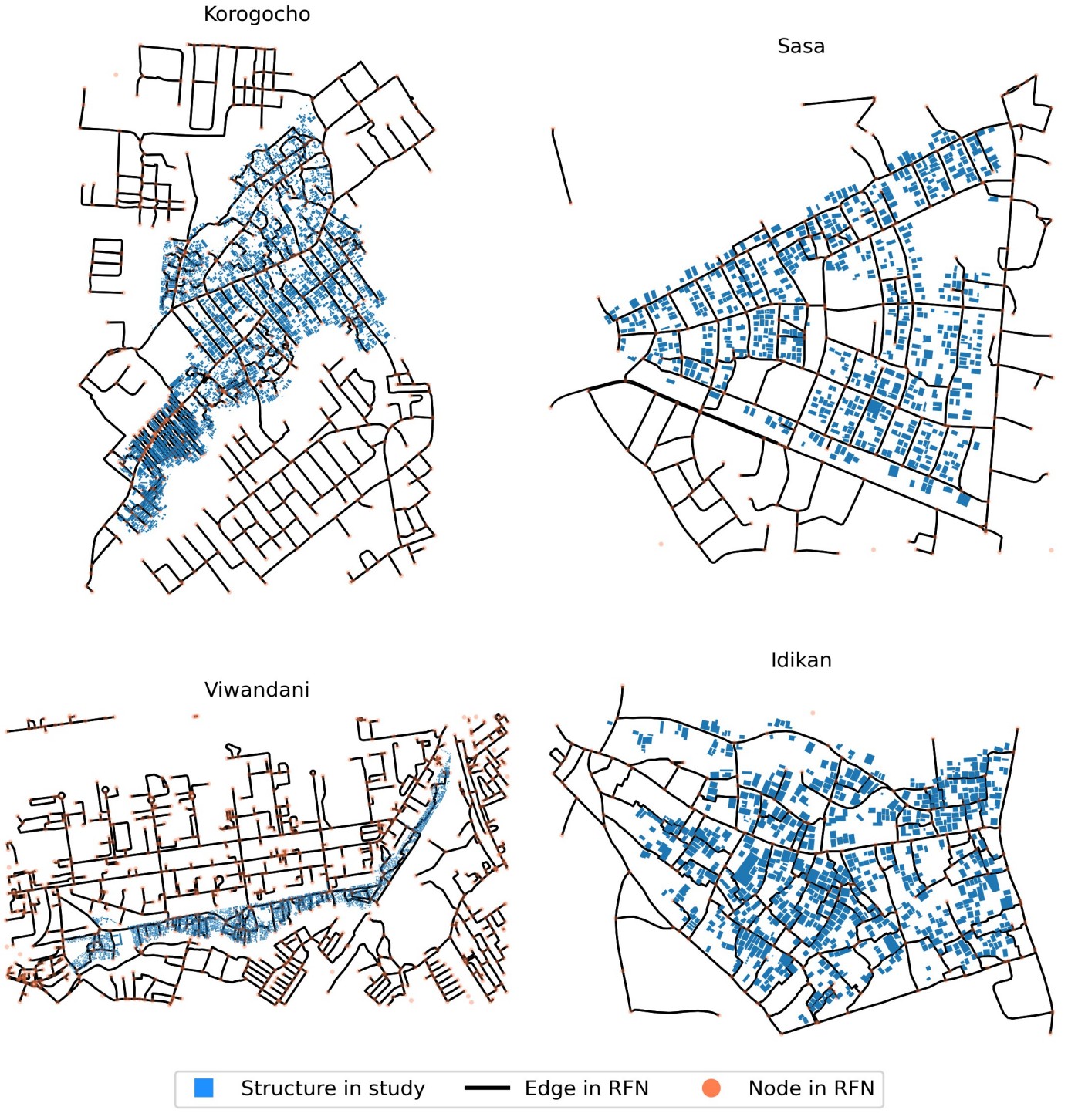

**Fig 2. Road and footpaths networks provided by OSM [20] overlaid with dwellings for each slum.**

granted full ethical approval by the University of Warwick Biomedical and Scientific Research Ethics Sub-Committee (REGO-2017-2043 AM01), the Ministry of Health, Lagos State Government (LSMH/2695/11/259), Research Ethics Committee of the Oyo State Ministry of Health (AD13/479/657), Amref Health Africa (AMREFESRC P440/2018), the National Bioethics Committee Pakistan (4-87/NBC-298/18/RDC3530) and the Bangladesh Medical Research Council (BMRC/NREC/2016-2019/759).

## Research approach

We describe our methods and their conceptual background here, and we note to the reader that the full formalizations, including supporting equations and detailed worked examples can be found in the supporting materials which we reference specifically throughout.

To determine the perceived quality of a facility, we use the notion of a 'bypass', a well-understood phenomenon in health-seeking behaviour, which indicates whether an individual has bypassed an HCP with a lower access cost to the trip's place of origin to visit another HCP with a greater access cost. Bypass behaviour denotes a choice (conscious or otherwise) made by an individual, which indicates that something about the visited HCP is worth incurring additional access costs (monetary and/or time) [12–15]. We assume that all trips originate at the individual's home (see below), and we calculate the access cost for the individual to all available HCPs. As we know which HCP the individual visited, we can then observe any bypasses as being those HCPs with a lower access cost than the visited HCP. We formalise the notion of bypassing in S5 Text.

The key concept behind this approach is that a recorded visit does not only contain information about the individual and the HCP they did attend, but also all other HCPs that they could have attended but did not. However, as we cannot determine the reason for these choices at an individual level, we aggregate them at the HCP level, and construct a 'perceived quality index' (PQI) of each HCP as a measure to quantify their perceived quality. The association between bypass behaviour and perceived quality is well known, Salazar et al. note "studies around the world have reported a link between patients' perceived quality of care and bypassing" [13].

To calculate the PQI, we start by assigning each HCP a PQI of 0. Then for each observed bypass, a negative contribution is made towards the bypassed HCP's PQI and a positive contribution made to the HCP the patient visited. The contribution is weighted based upon the observed magnitude of the difference between the access cost to the bypassed HCP and that to the visited HCP. The effect of this is to penalise to a greater extent those HCPs with lower access costs which could have been more easily visited, than those HCPs with a similar (but still lower) access cost to the visited HCP. This principle is adopted elsewhere, for example in two studies from China [22,23] which examine bypasses amongst hospital users. While neither of these studies seek to measure the perceived quality of a hospital, they both develop a bypass index to determine the propensity of patients to bypass other facilities based on travel costs, and similarly weight the distance of a bypass to the user. The formalisation of the PQI is detailed in S6 Text and a worked example is given.

To determine bypasses, we must determine the HCPs that each individual could, in fact, have visited instead. We therefore calculate the access cost for each individual to each HCP. The facility that was visited is recorded in the individual-level survey, so we can easily identify any facilities that have a lower access cost as those which were bypassed, and accordingly calculate the HCP-level PQI. Access costs are determined using the monetary cost of treatment (identified from the individual-level survey), the reported waiting time at the HCP, while travel time is estimated by calculating the shortest path in the RFN from the individual's home to the

HCP. We use a 'person-based' approach to calculate accessibility, which captures the trade-off a person will make between the costs to access a service and that service's perceived quality to the individual. Where sufficient data is available, person-based approaches that are based on aggregating individual observations of accessibility are preferable to other coarser grained methods such as place-based methods [24]. The full details of the access costs calculation are provided in S4 Text.

By using the access cost as the basis for the PQI calculations, we are controlling for the monetary cost and time cost (e.g., inconvenience) components of the HCP choices made by individuals, and are therefore calculating a residual term. This residual term captures the latent factors that pertain to HCP choice, such as reputation, familiarity, quality of customer service, etc. We note that this term does not necessarily relate to the quality of care provided by these facilities. This latent cost offsets any increase in access cost to visit a facility compared to those bypassed. We aggregate this cost at the HCP level to calculate the PQI. A high PQI denotes high levels of perceived quality.

Two key assumptions underpin our method. First, when an individual travels to an HCP, we assume that they travel from home, a common assumption in health-seeking studies [11,13,14]. This will not always be the case, but we demonstrate that a significant correlation exists between the reported travel times in the individual survey and the travel times we calculate (using homes as origin points), suggesting that this assumption is generally fair. We report these findings in S3 Text. The next assumption is that individuals always travel by foot. The assumption simplifies calculations and, while we know it will not always be the case, data from the individual surveys indicates that most trips (77.2%—see Table A in S1 Text) are carried out on foot.

## Statistical reporting

The raw data is processed and analysed using Python 3.9. The software depends on the following open-source packages: *osmnx*, *networkx*, *pandas*, *geopandas* and *numpy*, as well as native Python packages such as *math* and *statistics*. We provide a link to our code repository where the reader will also find details on how to run the script [25]. We do not make any assumptions about the underlying distributions that exist within the data, as our method is entirely data-driven (i.e., we do not depend on parametric methods). An exercise was carried out to remove non-sensical data, which was the result of either data entry errors (e.g., incorrect data types or values that are clearly not reasonable such as 999 minutes to travel to an HCP), or obvious 'place marker' values used by survey questioners. We encountered some missing data pertaining to the consultation cost and waiting time of a trip. In these cases, we use the mean observed value to the given facility collected for all other records for which missing data was not present. We do not consider a '0' (zero) value to be missing. Our analysis compares populations against one another to draw conclusion about HCP type and funding source, based on summary statistics such as the mean, median and standard deviation. Where we calculate the correlation between two populations, we are always referring to the Pearson correlation coefficient.

## Results

We present the results of our study in four parts. First, the popularity of each of the four HCP classes in each of the four slums and the access costs are examined. Second, we consider these two aspects together to identify whether a relationship between access costs and visits exists (e.g., do HCPs with low access costs receive high visitor numbers?). Third, we focus on observed bypasses within each slum to understand to what extent patients will bypass HCPs to visit HCPs further away. And finally, we present the PQI by HCP class to show which HCP

types have the highest perceived quality, and whether and to what extent patients are willing to incur higher costs to access.

## Popularity

Fig 3 shows the average number of visits to each of the HCP classes in each slum. When aggregated across the four slums, there is no HCP class that exhibits higher relative popularity, although some trends emerge within each slum. For Ibadan slums (Sasa and Idikan), public HCPs are more popular (accounting for 73% of all visits), while private HCPs are slightly more popular in the Nairobi slums (accounting for 53% of all visits). Private hospitals are more popular than public hospitals in the Nairobi slums (68% of hospital visits in Korogocho, and 54%

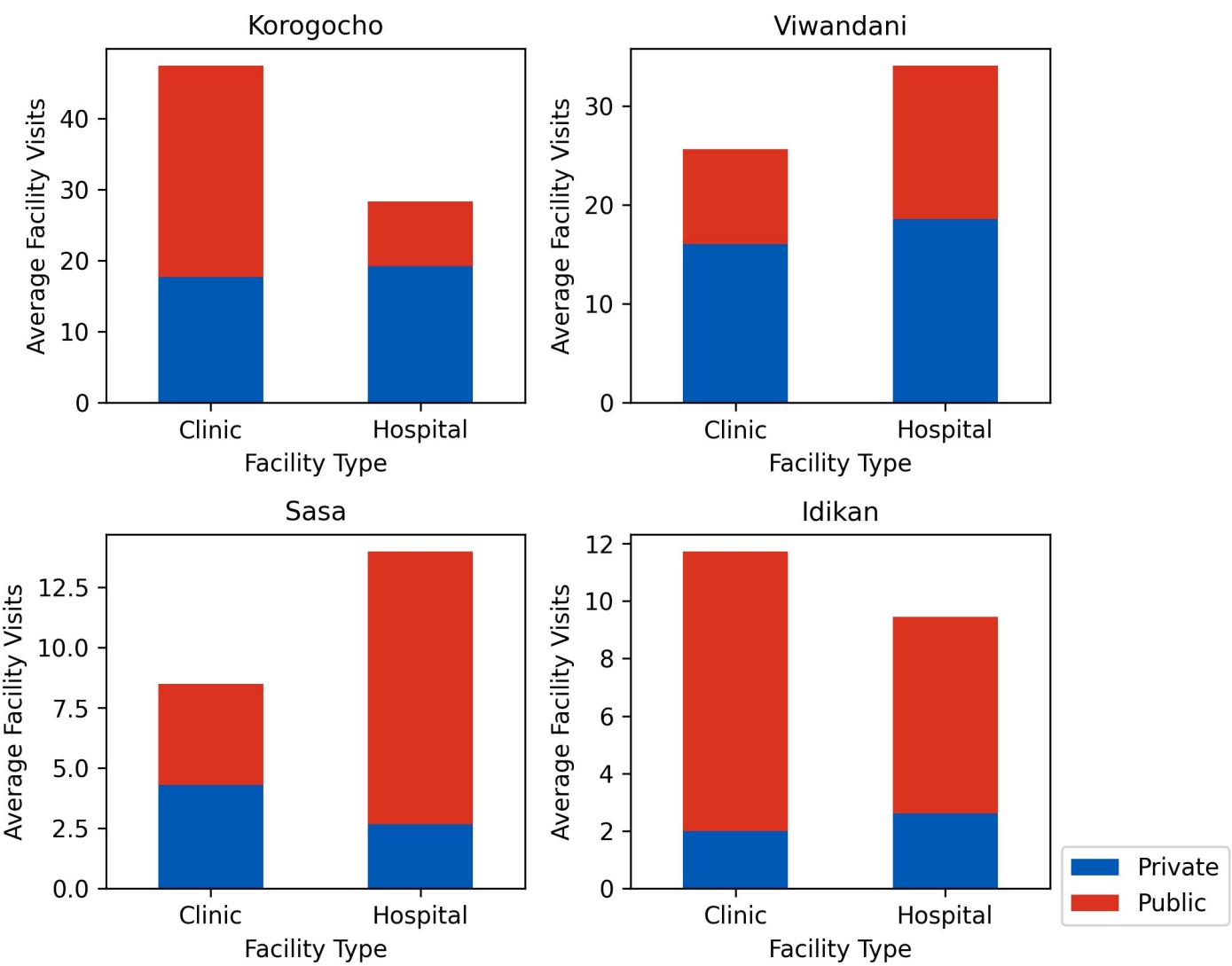

**Fig 3. Grouped bar chart showing average number of HCP visits by HCP type and funding source.**

in Viwandani), whereas in Ibadan, public hospitals are significantly more popular (81% of hospital visits in Sasa, and 72% in Idikan).

## Access costs

In Fig 4 we show the mean travel time and the mean access cost to each HCP class within each slum. For each slum, the mean travel time to a hospital is longer compared to clinics, with the difference in travel times often being quite significant (clinics are closer by 1.7 hours in Korogocho, 2.3 in Viwandani, 0.9 in Sasa, and 1.4 in Idikan). In most cases, the mean travel time to private HCPs is lower than to public facilities, with Korogocho being an exception. Moreover, travel time to private hospitals is lower compared to public hospitals, with the difference in travel times being significant in some cases (e.g., in Korogocho, the difference is 1.7 hours). These results indicate that clinics–especially the private ones–are focused on a more localised market, whereas hospitals–particularly public ones–attract patients from further away.

When *total* access costs are considered, including the cost associated to treatment, then the difference between private hospitals and public hospitals narrows and, in some places, private hospitals are now the most expensive option. Overall, the access cost to hospitals is always higher than clinics.

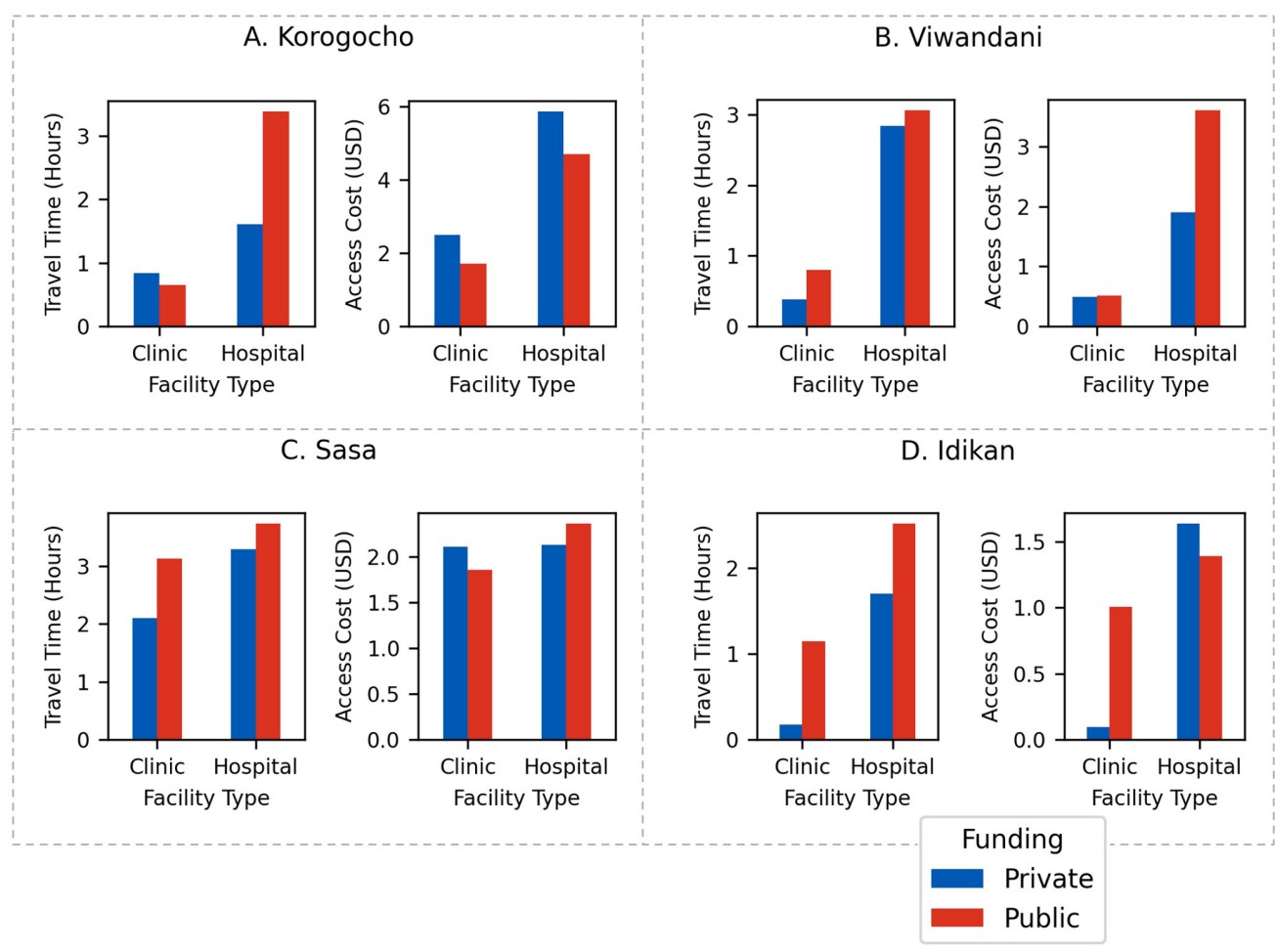

**Fig 4. Bar plots for each slum showing the mean travel time and the mean access cost to each of the four HCP classes.**

## Access costs and popularity

In this section, we present the results corresponding to the relationship between popularity and access cost. If access costs were the sole determinant in HCP choice, we would expect a strong negative relationship between these features (i.e., low access costs result in high visitor numbers). Fig 5 shows that, while some negative correlation exists, the relationship is not strong. From the negligible correlation coefficients (shown in Fig 5), it is evident that the reasons for HCP choice extends beyond just the access costs, and that perceived quality trades off against net costs.

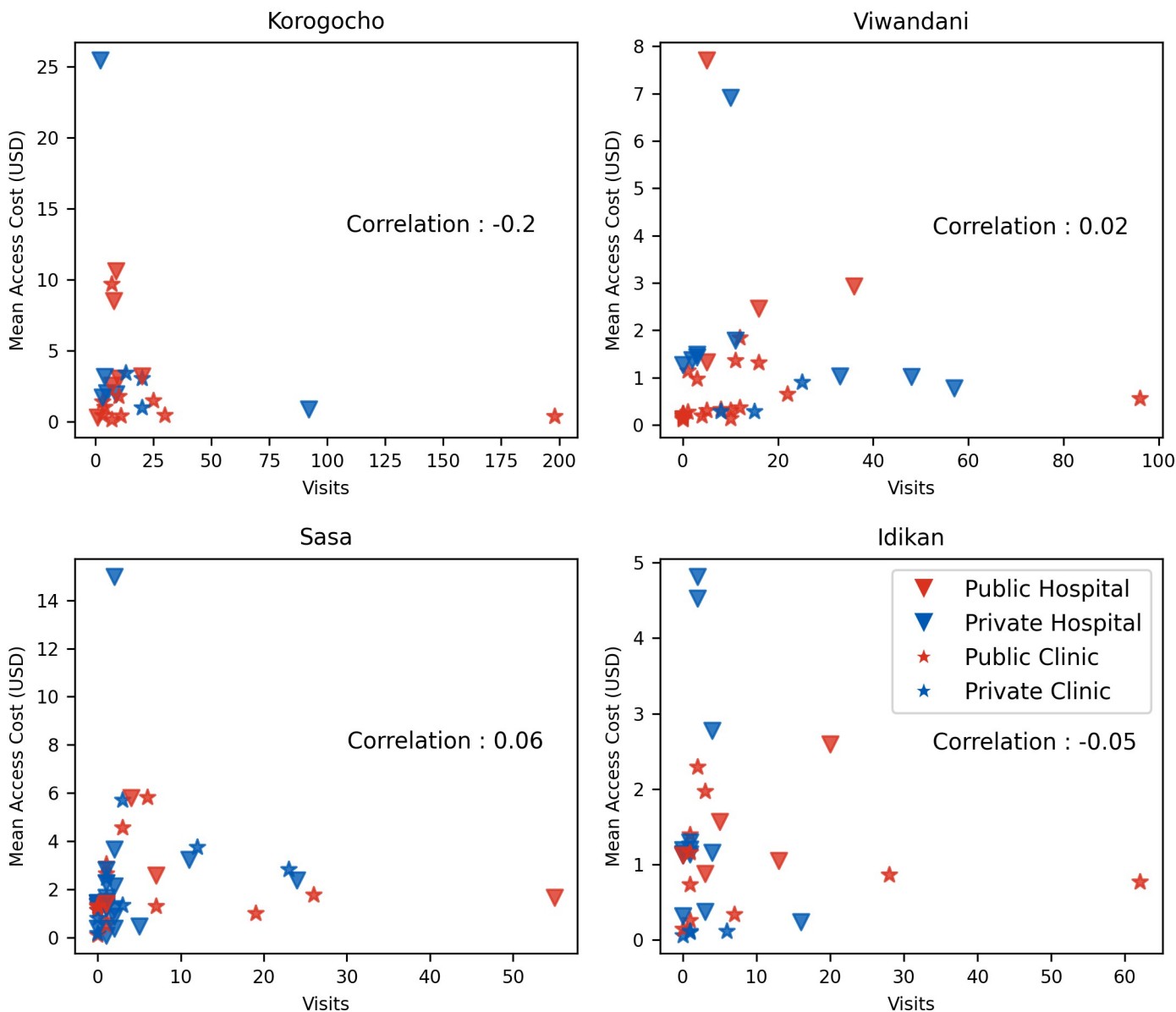

**Fig 5. Scatter plot of HCPs in each slum showing recorded visit number against access cost with Pearson correlation coefficient.**

## Examining bypass behaviour

The PQI aggregates bypass behaviour across observed HCP visits. In this section, the extent of bypass behaviour and associated patterns are investigated. Fig 6 shows, for each slum, the average number of times that each HCP class is bypassed, and the average number of facilities that are bypassed per visit to each HCP class. These results highlight the extent of bypassing that occurs, e.g., many HCPs are bypassed by hundreds of people in favour of alternative HCPs, and for the average visit to an HCP often dozens of other HCPs are bypassed. Clinics tend to be bypassed more frequently than hospitals, and private facilities tend to be bypassed more frequently than their public counterparts. In all cases, public hospitals are bypassed the least often.

When considering the number of facilities bypassed per visit, we can observe that clinics are bypassed far more frequently than hospitals–there are 114% more bypasses of clinics than hospitals across all slums. However, private and public HCPs are bypassed a similar number of times across all slums and HCP types. Publicly funded hospitals are the HCPs with the lowest number of bypasses. Overall, more bypasses are made when visiting a hospital compared to visiting a clinic (19.6 vs. 10), and more bypasses occur when visiting a public HCP compared to a private HCP (17.7 vs. 11.9). When someone visits a public hospital, they bypass 23.2 other facilities on average: the highest of any facility class.

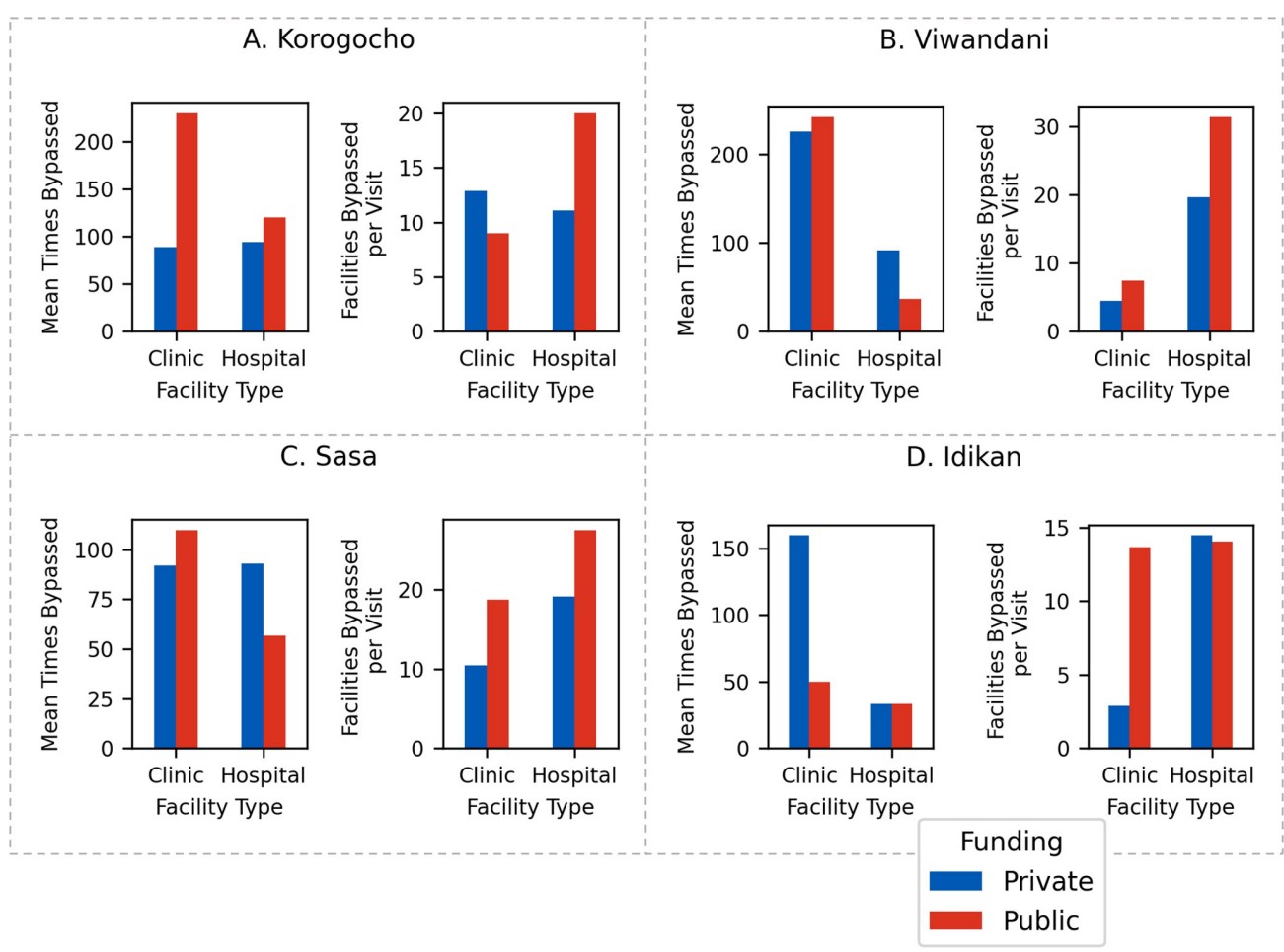

**Fig 6. Bar charts demonstrating bypass behaviour in each slum, showing the mean number of times each HCP class is bypassed, and per visit how many other HCPs are bypassed by HCP class.**

### Analysing the perceied quality index

In this section, we analyse how the PQI varies across our HCP classes, as illustrated in Fig 7. Overall, hospitals tend to have a higher PQI compared to clinics, and publicly funded HCPs tend to have a higher PQI compared to privately funded HCPs. In all slums, publicly funded hospitals have the highest perceived quality among the four HCP classes. While there is a clear preference for public services within the hospital class, this preference is less clear for clinics.

Fig 8 shows the distribution of the PQI. An important observation is that, in each slum, the HCP with the highest PQI is always a public hospital. In some cases, such as Idikan and Sasa, a significant difference from the HCP with highest PQI to the next one is observed. Table 2 shows the mean PQI across all slums. Hospitals tend to have a higher PQI than clinics. Publicly

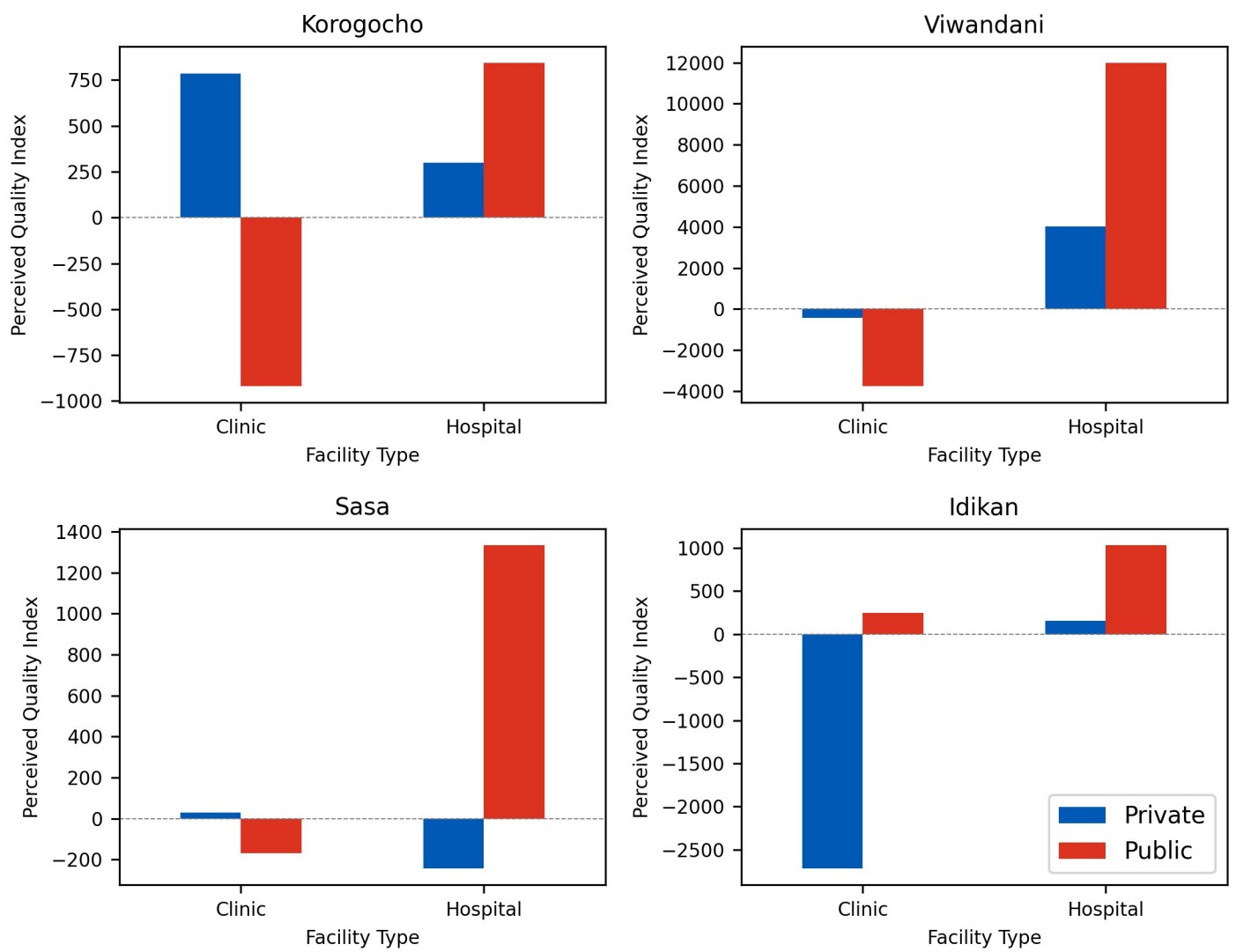

**Fig 7. Mean HCP attractiveness index score.**

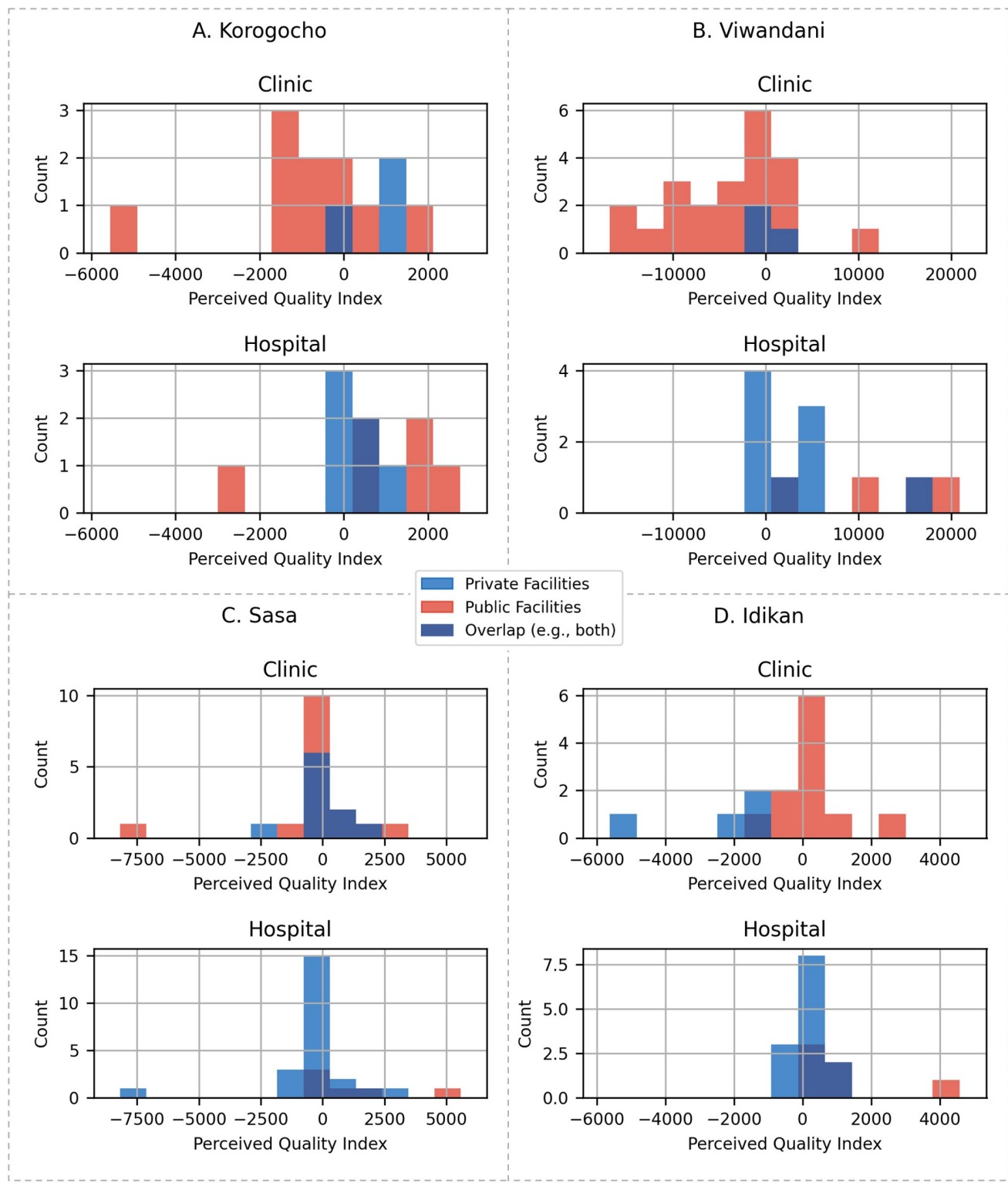

**Fig 8. Histogram of distribution of PQI for HCPs in each slum.**

**Table 2. Mean attractiveness index for all HCPs across all slums.**

| HCP Type | Privately Funded | Publicly Funded | All |
|---|---|---|---|
| Clinic | -585 | -1152 | -868 |
| Hospital | 1056 | 3800 | 2428 |
| All | 235 | 1324 | 780 |

funded hospitals are the HCPs with the highest PQI across all slums, and by quite a significant margin. In Table 3, the mean standard deviation of the PQI over all slums is shown. This demonstrates how much variability exists between HCPs within each of the facility classes. Hospitals, particularly if publicly funded, exhibit high levels of variability in their reported PQI. Considering that public hospitals generally report higher PQI scores, given the high variability within this category, it is likely that this is being driven by the small number of public hospitals that have a high PQI.

Fig 9 shows the PQI against the mean access cost. The correlation is positive for all slums and strongly positive for some.

## Discussion

From our results we can conclude that the most attractive HCPs are often those that are the least accessible (i.e., that people will incur additional cost/inconvenience to visit these facilities). This suggests that the more localised network of HCPs, which tend to comprise smaller, less formal practices, are being crowded out by larger facilities further away. Here we discuss these findings in the context of wider literature, the conservative approach we have taken when considering the study's limitations and the wider policy implications of our findings.

### The 'crowding out' hypothesis

In their summary of the Lancet Series on UHC and Private Health Care, Hanson and McPake [8] postulate that "a publicly financed health service can crowd out the low-quality element of the private sector". This paper has investigated the association between patient choice and the type of healthcare facility in four slums in Nigeria and Kenya. The results provide evidence for the 'crowding out' hypothesis, indicating that people will bypass local, small-scale HCPs to reach larger-scale hospitals much more often than the reverse. Amoro et al. note similar findings when investigating maternity service bypasses in LMICs–stating "[w]hat is clear is that in LMICs, it is very common for healthcare seekers to move straight to the district, regional and teaching hospitals to seek care without any point of contact with their PHC [primary health care] facilities" [26]. We investigate this further through the application of a generalised access cost to determine bypasses and the perceived quality at the HCP level. Thus considering the net out-of-pocket monetary costs and inconvenience factor (e.g., time) to reach a facility, we find that hospitals are preferred on average despite greater distance and cost. When it comes to the difference between public and private hospitals, we found that public hospitals had a higher PQI than private hospitals. The situation regarding clinics, however, was more mixed.

**Table 3. Mean standard deviation of AI for all HCPs across all slums.**

| HCP Type | Privately Funded | Publicly Funded | All |
|---|---|---|---|
| Clinic | 1428 | 2890 | 2159 |
| Hospital | 1990 | 3659 | 2824 |
| All | 235 | 1324 | 2492 |

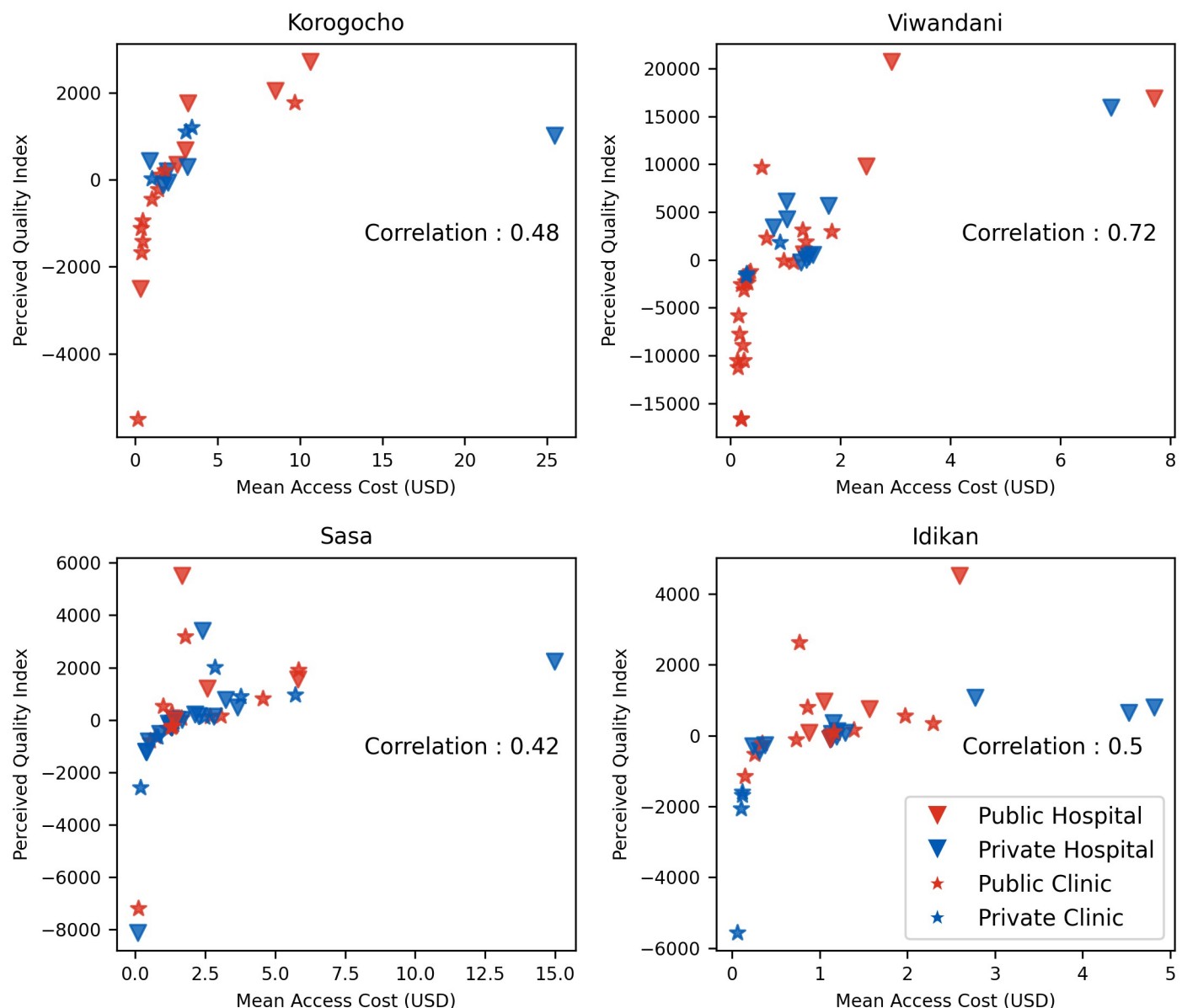

**Fig 9. Scatter plot between attractiveness index and access cost with Pearson correlation coefficient.**

While much work exists on the phenomena of bypassing, these results present, to our knowledge for the first time, data supporting the 'crowding out' hypothesis by tracking individual behaviours as opposed to public health spending in aggregate. We cannot, of course, prove that facilities closed as a result of expressed preferences, but it is evident that people are discriminating in their behaviour. This discrimination encompasses the most attractive class of HCP–even among public hospitals, there was a large variation in attractiveness. Our findings, however, relate to perceived quality and do, not represent an independent measure of technical aspects of quality, such as making the correct diagnosis or prescribing appropriately.

## Quality of care

In our study we did not measure quality of care in terms of its technical features, however Sabde et al. investigated whether bypassing maternity services in India can be explained by the technical quality of care (number of doctors, drug availability, equipment etc) and discovered a statistically significant association [27]. Moreover, there is a large literature on the quality of ambulatory care in poor LMIC communities by small (often single-handed) providers, which shows that it is generally of poor quality, whether provided by doctors or other cadres [28–33]. In contrast, a study from a large public hospital in Nigeria shows high quality care in the general outpatient clinic [34]. Thus, such limited evidence suggests that patients may be discerning in their choice of HCP–market failure is common in healthcare, but this does not mean that there is no correlation between perceived quality and technical quality. Moreover, when it comes to communication and patient-centred care, we must allow patients to be the best judge. It is clear from our findings that patients exhibit preferences, and that these favour larger public hospitals, as predicted in the Lancet Series.

## Limitations

**Health-seeking behaviour is not uniform across different reasons for seeking care.** The perceived severity of a condition will influence choice of HCP; the less serious a symptom is perceived to be, the less a person will be willing to trade cost or convenience to ascertain higher perceived quality care [11]. A person with angina pain is more likely to tolerate inconvenience than one with heartburn, or indeed someone's condition may be so severe as to require hospital care. However, this 'bias' dilutes our findings since it must be assumed that the less severe the condition, the less the inconvenience that will be tolerated. In other words, under our premise, even more people would have bypassed HCPs close at hand to reach a hospital had we been able to stratify our data according to the reported severity of symptoms.

**A visit might originate at home.** Some HCP visits might originate from places other than a person's home–for example, from the place of work. To support our findings, we demonstrate that our assumption around trip origin holds true in most cases; we calculated the correlation between the reported travel time of the individual (taken from the survey data) and the calculated travel time from the individual's home to the chosen HCP (which is used in the calculation of the access cost). We report strong positive correlations in S3 Text. In our previous research [5], we find that there is a significantly longer average time to visit hospitals as opposed to clinics, which we also observe here. Several other studies looking into health-seeking behaviour also make this assumption [11,13,14].

**A patient may bypass a facility because they did not know it was there.** It is possible that a clinic could be bypassed because the person seeking care does not know it was there. However, while this may happen on occasions, it is unlikely to explain the extent and frequency with which bypasses occurred. People arriving at hospital will have bypassed numerous clinics and cannot have been ignorant of them all. Informal settlements have many clinics, so people attending a hospital are unlikely to have been unaware of all of the clinics close to their route. People living in slums are likely to share knowledge, and it must be assumed that people responsible for clinics are wishing to attract custom.

**Mode of transport may vary.** We assume all trips are carried out on foot, but it is intuitive to think that longer trips are more likely to use some form of transportation for at least part of the trip. To support this assumption, we report in Table A in S1 Text that 77% of the reported trips in the survey were taken on foot. The next largest category is public transport (18%). We did not model the public transport component as we do not have enough granularity to do so accurately (e.g., there are multiple modes of public transport, journeys are likely to be a mix of

modes), and obtaining detailed, accurate data of the public transportation infrastructures in LMICs is hard. This may not significantly impact the calculated access costs, as the monetary cost would increase even if journey time would decrease.

**Implications for policy.** Policy responses to the finding that people prefer–and are probably right to prefer–hospital services over more dispersed and informal providers could theoretically be of two types. The response could be to favour and develop hospitals or to strengthen local provision. However, resources are severely constrained in LMIC cities, and are likely to remain so in Africa where populations are growing as fast (or faster) than the economy as a whole [2]. While we are not arguing that international policy should turn on our single study, we think that our findings, that people place a premium on hospital outpatient services over the variegated smaller clinics in close proximity to their homes, could have provocative policy implications. Under limited budgets policy makers could either try to improve a large number of small providers or concentrate resources on a smaller number of larger providers. The WHO Astana declaration on primary care [35] could be interpreted as advice in favour of the first option. However, the declaration is careful not to conflate primary care with non-hospital care. Many general outpatient clinics are staffed by primary care providers. Moreover, the data show that in LMIC cities there is a profusion of suppliers even in informal settlements/slums such that access to services is not the main issue; rather the quality of services is now the limiting factor. Our results are relevant to the issue; taken in the round with data on quality of care [26] they support a move to incentivise some consolidation of services. Centralisation of services may be complemented by a hub and spoke model. Such a model could include integration of pharmacy services since previous research in LMICs [36,37] and our previous work in informal settlements [21] shows that, while there are very large numbers of pharmacies, pharmacy provision is rudimentary. The Alma-Ata Declaration of 1978 strongly supported the development of primary care and argued that it should have more priority for resources [38]. We suggest that a more nuanced approach, to the balance of hospital and local care is in order and argue that hospital-directed care should be regarded as part of the solution, rather than part of the problem. Such a solution is further supported by the Lancet series finding that reasonable quality public services may not only 'crowd out' informal providers, but that they may drive up the quality of such small and private providers that remain in the market. Thus, as countries extend the scale and coverage of universal health coverage, they can use their new purchasing power to strengthen those parts of the system for which patients already show some predilection.

## Implications for research

The revealed preferences in this study would be complemented by studies of hypothetical choices to dissect in finer detail which attributes of different providers were most salient to local people. We are currently analysing such data from Ibadan and the Azam Basti informal settlement in Karachi, Pakistan. Studies of the technical quality of care and of barriers and facilitators to improved care are also required as an urgent matter across multiple HCPs in LMIC cities. Finally, a more bespoke data collection could be carried out with the purpose of access cost calculation. This would address some of the limitations discussed above (such as mode of transport, and the 'travel from home' assumption), as well as provide more detailed information at HCP level (providing more accurate waiting time, medicine costs, etc). More detailed demographic information about the underlying population could also benefit further studies in this area.

## Conclusion

This study has sought to provide evidence for the 'crowding out' hypothesis, which states that in dense urban areas in LMICs, and particularly slum areas, more established healthcare facilities

will crowd out the smaller more informal network of providers. To establish this, we formalised the notion of 'perceived quality'. First, we observed the monetary cost and inconvenience (time cost) related to an individual's choice of HCP, and then stated that any residual cost is therefore the perceived quality associated to that HCP. By calculating the access cost pertaining to individual HCP visits (a combination of monetary costs and inconvenience) and observing bypasses relating to these visits, we aggregated this residual cost at the HCP level, and call this the perceived quality index. We have shown that publicly funded hospitals, which tend to be further away, have, on average, the highest levels of perceived quality. Even within this category, there is much variation, suggesting that only a small number of facilities benefit from this 'perceived quality', and that they do indeed crowd out other providers. The policy implications of this study are to continue to support the development of centralised services for primary care, particularly publicly funded hospitals, until the quality of more local providers can be assured. This is particularly the case given that evidence suggests that the 'crowding out' hypothesis, which we have demonstrated, can itself drive up the quality of local providers.

## Supporting information

**S1 Fig.**
(TIF)

**S1 Text. Assumption that journey are carried out on foot.**
(DOCX)

**S2 Text. Assumption that clinics and hospital are comparable.**
(DOCX)

**S3 Text. Assumption that journeys originate from people's homes.**
(DOCX)

**S4 Text. Defining the access cost.**
(DOCX)

**S5 Text. Defining bypass behaviour.**
(DOCX)

**S6 Text. Calculating the attractiveness index.**
(DOCX)

**S7 Text. Implementation details.**
(DOCX)

## Acknowledgments

We acknowledge the contributions of the NIHR Global Health Research Unit on Improving Health in Slums teams based at the University of Ibadan, Nigeria and the African Population and Health Research Center, Kenya. Particularly the work of Caroline Kabaria, Akinyinka Omigbodun and Motunrayo Ayobola.

## Author Contributions

**Conceptualization:** Chris Conlan, Teddy Cunningham, Sam Watson, Jason Madan, Hakan Ferhatosmanoglu, Richard Lilford.

**Data curation:** Chris Conlan, Teddy Cunningham, Sam Watson.

**Formal analysis:** Chris Conlan, Teddy Cunningham, Sam Watson, Hakan Ferhatosmanoglu, Richard Lilford.

**Funding acquisition:** Jo Sartori, Richard Lilford.

**Investigation:** Chris Conlan, Jason Madan, Hakan Ferhatosmanoglu, Richard Lilford.

**Methodology:** Chris Conlan, Teddy Cunningham, Sam Watson, Jason Madan, Hakan Ferhatosmanoglu, Richard Lilford.

**Project administration:** Chris Conlan, Teddy Cunningham, Jo Sartori.

**Resources:** Jo Sartori.

**Software:** Chris Conlan, Hakan Ferhatosmanoglu.

**Supervision:** Jason Madan, Hakan Ferhatosmanoglu, Richard Lilford.

**Validation:** Chris Conlan, Teddy Cunningham.

**Visualization:** Chris Conlan.

**Writing – original draft:** Chris Conlan, Teddy Cunningham, Alexandros Sfyridis, Richard Lilford.

**Writing – review & editing:** Chris Conlan, Teddy Cunningham, Sam Watson, Jason Madan, Alexandros Sfyridis, Jo Sartori, Hakan Ferhatosmanoglu, Richard Lilford.

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
