## [Decision Letter · Decision Letter 0]

1 Sep 2022

PGPH-D-22-01013

Perceived quality of care and choice of healthcare provider in informal settlements

Dear Dr. Conlan,

Thank you for submitting your manuscript to PLOS Global Public Health. After careful consideration, we feel that it has merit but does not fully meet PLOS Global Public Health’s publication criteria as it currently stands. Therefore, we invite you to submit a revised version of the manuscript that addresses the points raised during the review process.

We look forward to receiving your revised manuscript.

Kind regards,

Anteneh Asefa Mekonnen

Academic Editor

Journal Requirements:

1. Please amend your online detailed Financial Disclosure statement. This is published with the article. It must therefore be completed in full sentences and contain the exact wording you wish to be published.

2. Please ensure that the funders and grant numbers match between the Financial Disclosure field and the Funding Information tab in your submission form. Note that the funders must be provided in the same order in both places as well.

3. Please update your online Competing Interests statement. If you have no competing interests to declare, please state: “The authors have declared that no competing interests exist.”

4. We do not publish any copyright or trademark symbols that usually accompany proprietary names, eg (R), (C), or TM  (e.g. next to drug or reagent names). Please remove all instances of trademark/copyright symbols throughout the text, including © on page 21.

Additional Editor Comments (if provided):

Reviewers' comments:

Reviewer's Responses to Questions

**Comments to the Author**

1. Does this manuscript meet PLOS Global Public Health’s publication criteria? Is the manuscript technically sound, and do the data support the conclusions? The manuscript must describe methodologically and ethically rigorous research with conclusions that are appropriately drawn based on the data presented.

Reviewer #1: Yes

Reviewer #2: Yes

2. Has the statistical analysis been performed appropriately and rigorously?

Reviewer #1: I don't know

Reviewer #2: Yes

3. Have the authors made all data underlying the findings in their manuscript fully available (please refer to the Data Availability Statement at the start of the manuscript PDF file)?

Reviewer #1: Yes

Reviewer #2: Yes

4. Is the manuscript presented in an intelligible fashion and written in standard English?

Reviewer #1: Yes

Reviewer #2: Yes

5. Review Comments to the Author

Reviewer #1: Interesting ms...it runs counter to what we experience in LMICs e.g., in Bangladesh (Ref: http://www.health-policy-systems.com/content/5/1/1;
http://www.biomedcentral.com/1472-6963/11/S2/S8; HEALTH POLICY AND PLANNING LANNING;15(1):95-102 )...the current findings need to be discussed in the context of findings from e.g., LMICs in Asia (relevant literature is conspicuously absent)!!

Second, the ms need to be examined critically for statistical calculations of variables such as 'access cost', 'attractiveness index', 'perceived quality', 'bypass behaviour' etc. with appropriate conceptual background and appropriate references! not clear if the QoC is better in Kemya/Nigeria than in BD (ref please)!

In the limitations, quite a few critical confounder is mentioned...despite these, the argument of the paper/validity of the data should be questioned...isn't the conclusions a little too ambitious?

'If the quality gradient suggested by the Lancet series and by our findings is confirmed, then the development of services should not be tilted in favour of smaller clinics, until quality can be better assured among such providers'. A dangerous policy proposition for the LMICs across the worls, in direct opposition to the theme of renovated PHC for UHC Call (Astana 2018)!

In short, the ms need to be positioned in the context of wider literature on the opic, not only pro but also opposite views to acept the conclusions and policy implications unreserved

Reviewer #2: The manuscript is well written. I have just the following few comments:

1. A key should be provided for Figure 2

2. Figure 3 and Figure 4 are not very clear. In fact, the quality of the figures can be improved. The units of measurement should be indicated in the figures

3. There is a lot of commentary in the presentation of the results. It is good that the authors should reduce the commentary and try to go straight forward to present the results.

4. The discussion section should be expanded to ensure that the evidence of this current study is compared with findings from similar studies in other low and middle income countries such https://bmchealthservres.biomedcentral.com/articles/10.1186/s12913-021-06573-3 -Bypassing primary healthcare facilities for maternal healthcare in North West Ghana: socio-economic correlates and financial implications

6. PLOS authors have the option to publish the peer review history of their article (what does this mean?). If published, this will include your full peer review and any attached files.

**Do you want your identity to be public for this peer review?** For information about this choice, including consent withdrawal, please see our Privacy Policy.

Reviewer #1: **Yes: **Syed Masud Ahmed

Reviewer #2: No

---

## [Decision Letter · Decision Letter 1]

19 Jan 2023

Perceived quality of care and choice of healthcare provider in informal settlements

PGPH-D-22-01013R1

Dear Mr Conlan,

We are pleased to inform you that your manuscript 'Perceived quality of care and choice of healthcare provider in informal settlements' has been provisionally accepted for publication in PLOS Global Public Health.

Best regards,

Hassan Haghparast Bidgoli

Academic Editor

Thank you for addressing the comments raised by the reviewers. There is no further comment from the reviewers and the editor.

Reviewer Comments (if any, and for reference):

Reviewer's Responses to Questions

**Comments to the Author**

1. If the authors have adequately addressed your comments raised in a previous round of review and you feel that this manuscript is now acceptable for publication, you may indicate that here to bypass the “Comments to the Author” section, enter your conflict of interest statement in the “Confidential to Editor” section, and submit your "Accept" recommendation.

Reviewer #1: All comments have been addressed

2. Does this manuscript meet PLOS Global Public Health’s publication criteria? Is the manuscript technically sound, and do the data support the conclusions? The manuscript must describe methodologically and ethically rigorous research with conclusions that are appropriately drawn based on the data presented.

Reviewer #1: Yes

3. Has the statistical analysis been performed appropriately and rigorously?

Reviewer #1: Yes

4. Have the authors made all data underlying the findings in their manuscript fully available (please refer to the Data Availability Statement at the start of the manuscript PDF file)?

Reviewer #1: Yes

5. Is the manuscript presented in an intelligible fashion and written in standard English?

Reviewer #1: Yes

6. Review Comments to the Author

Reviewer #1: No more comments, thnaks!

7. PLOS authors have the option to publish the peer review history of their article (what does this mean?). If published, this will include your full peer review and any attached files.

**Do you want your identity to be public for this peer review?** For information about this choice, including consent withdrawal, please see our Privacy Policy.

Reviewer #1: **Yes: **Prof. Syed Masud Ahmed, BRAC JPG School of Public Health, BRAC University, Bangladesh.
